# Is the Intergenic Region of *Aedes aegypti* Totivirus a Recombination Hotspot?

**DOI:** 10.3390/v14112467

**Published:** 2022-11-08

**Authors:** Roseane da Silva Couto, Geovani de Oliveira Ribeiro, Ramendra Pati Pandey, Élcio Leal

**Affiliations:** 1Laboratório de Diversidade Viral, Instituto de Ciências Biológicas, Universidade Federal do Pará, Belem 66075-000, Pará, Brazil; 2Centre for Drug Design Discovery and Development (C4D), SRM University, Delhi-NCR, Rajiv Gandhi Education City, Sonepat 131029, Haryana, India

**Keywords:** recombination, totivirus, *Aedes*, insect, RNA-polymerase, hairpin structure, insect viruses

## Abstract

The genus *totivirus* in the family *Totiviridae* contains double-stranded RNA viruses. Their genome has two open reading frames (ORFs) that encode capsid protein (CP) and RNA-dependent RNA polymerase (RdRp). The toti-like viruses recently identified in *Anopheles* sp. and *Aedes aegypti* mosquitoes (AaTV) share the same genome organization as other totiviruses. The AaTVs that have been described in distinct geographical regions are monophyletic. In this study, we show that AaTV sequences can be grouped into at least three phylogenetic clades (named A, B, and C). Clades A and B are composed of AaTV sequences from mosquitoes collected in the Caribbean region (Guadeloupe), and clade C contains sequences from the USA. These clades may represent AaTV lineages that are locally adapted to their host populations. We also identified three recombinant AaTV strains circulating in mosquitoes in Guadeloupe. Although these strains have different chimeric patterns, the position of the recombination breakpoint was identical in all strains. Interestingly, this breakpoint is located in a hairpin-like structure in the intergenic region of the AaTV genome. This RNA structure may stall RNA polymerase processivity and consequently induce template switching. In vitro studies should be conducted to further investigate the biological significance of AaTV’s intergenic region as a recombination hotspot

## 1. Introduction

The family *Totiviridae* contains five genera (i.e., *Giardiavirus*, *Leishmaniavirus*, *Totivirus*, *Trichomonasvirus*, and *Victorivirus*) [1,2]. The genus *Totivirus* contains viruses initially identified in fungi [3]. More toti-like viruses have been found by NGS in developed hosts in recent years [4,5,6,7]. The majority of these have been discovered in a variety of arthropods, including the fruit fly, mosquito, ants, and ticks [4,8,9,10,11,12], the seawater-dwelling arthropods shrimp and horseshoe crab [13], as well as roundworms and flatworms, bats [14] (i.e., in feces, which have been linked to insectivorous bats) [9], and the two fish species Atlantic salmon (*Salmo salmo*) [15]. The toti-like viruses found in the mosquito’s host (*Anopheles* and *Aedes aegypti*) share the genomic characteristics described for *Totiviridae* and share an ancestral monophyletic ancestor [10,16,17,18]. These viruses have been tentatively placed in the *Totiviridae* family and have not been definitively classified. Likewise, Guadeloupe *Aedes aegypti* totivirus (AaTV) is an insect-specific virus that was identified in *Aedes aegypti* from the Caribbean [10] and California (United States) [17]. They have a genome of 4700–6700 nucleotides in length, and the genome contains two large overlapping putative open reading frames (ORFs) [8,19,20,21]. The 5′-proximal ORF encodes the capsid protein (Cap), and the 3′-proximal ORF encodes an RNA-dependent RNA polymerase (RdRp). The 5’ end of the positive strand of the dsRNA genome has no cap and is very structured. These viruses contain a long 5’ untranslated region (5’ UTR), which functions as an internal ribosome entry site (IRES). They also have noncoding regions between the ORFs [1,22].

In a previous study, the authors speculated about a possible recombination event, but the results were inconclusive [10]. Here, we explored in detail the recombination patterns in the *Aedes aegypti* totivirus (AaTV). We found three sequences with similar mosaic patterns. They also share the location of the single breakpoint detected in the intergenic region of the AaTV. We also hypothesized that a predicted secondary RNA structure found in the intergenic region of AaTV may halt the processivity of RNA-polymerase. This can induce the RNA-polymerase to detach and jump to another RNA target template, thus increasing the chances of genomic recombination.

## 2. Materials and Methods

### 2.1. Dataset

We downloaded all 25 near-complete genomes of *Aedes aegypti* totivirus (AaTV) from NCBI (https://www.ncbi.nlm.nih.gov/nuccore/?term=Aedes+aegypti+totivirus, accessed on 22 September 2022). We also included in this dataset one Totivirus sequence detected in Fungi (MZ868719), and Totiviruses sequences of other insects (MZ209742, KX148550, MN661076, and MW520394). 

### 2.2. Alignment and Phylogenetic Analysis

The alignment was performed using MAFFT [23] and edited using AliView [24]. An initial tree was inferred using the maximum likelihood criterion, and branch support values were evaluated using the Shimodaira–Hasegawa test. The likelihood ratio test (LRT), which is a feature of the jModeltest software [25], was used to determine the best-fit evolutionary model for each tree. All maximum likelihood trees were constructed using PhyML [26]. The software SplitsTree V.5 [27] was used to construct a phylogenetic network to show that AaTV sequences split into three clades. We utilized a pair-wise method implemented in the program SDT v. 1.2 [28] to estimate sequence similarity scores. Algorithms implemented in MUSCLE were used to estimate the similarity alignments of each unique pair of sequences [29]. The program then utilizes the NEIGHBOR component of PHYLIP to construct a tree [30] after computing the identity score for each pair of sequences. All sequences were arranged in the rooted neighbor-joining phylogenetic tree according to their apparent evolutionary relatedness. The results are displayed as a frequency distribution of the pairwise identity score matrix.

### 2.3. Recombination Signal

We used RDP v.5 software [31], which utilizes a collection of methods to determine the extent of recombination in AaTV sequences. A brief description of these methods follows, and an excellent and detailed explanation of each method implemented in the RDP program can be found in the user’s manual (available online: http://darwin.uvigo.es/rdp/rdp.html/ accessed on 22 September 2022). 

### 2.4. RNA Secondary Structure Prediction

To predict the secondary RNA structure, we used the RNAfold method of the Vienna RNA web services (available at http://rna.tbi.univie.ac.at/cgi-bin/RNAWebSuite/RNAfold.cgi/ accessed on 22 September 2022). RNAfold predicts the minimum free energy (mfe) structure of a single sequence using the classic algorithm of Zuker and Stiegler, which uses thermodynamics and auxiliary information. In addition, the method calculates equilibrium base pairing probabilities via John McCaskill’s partition function algorithm.

## 3. Results

### 3.1. Phylogenetic Analysis of Near-Complete Genomes of Totiviruses

Initially, we inferred a maximum likelihood tree using all available AaTV sequences and some other totiviruses as references. This tree shows that all AaTVs are monophyletic (Figure 1). In addition, the short branch in the clade formed by all AaTVs indicates that they have radiated recently. It is interesting to note that the sequences of totiviruses identified in *Aedes aegypti* (AaTVs) and one totivirus (LC496074) identified in *Anopheles gambiae* (indicated by the arrow in the tree of Figure 1) are very divergent. The genetic divergence between the sequence LC496074 and AaTV sequences was higher than 39%, while the mean divergence among all AaTV sequences was 6%.

### 3.2. Phyloclades of AaTV

Owing to the observed low divergence among AaTV sequences, we were interested in determining whether AaTV sequences could be classified into distinct groups. We used the split decomposition method, assuming the KHY85 evolutionary model, to construct a phylogenetic network that shows that AaTV can be grouped into at least three clades (clades A, B, and C in Figure 2a). Clades A and C were composed of sequences from Guadeloupe, France, and Clade C was composed of AaTV isolated from the USA. In addition, we showed that AaTV can be clearly demarcated into distinct groups based on the similarity scores of sequences. The identity matrix (Figure 2b) also indicates that sequences belonging to these clades have similarity scores higher than 98%.

### 3.3. Topologies of Capsid and RNA-Polymerase Trees of AaTV

It is known that structural genes such as capsid have higher evolutionary rates compared to structural genes such as RNA-dependent RNA-polymerase (RdRp). Thus, we were interested in determining the extent to which varying substitution rates affected the clade composition detected in our previous analysis. To minimize the effect of rate heterogeneity in the inferences, we used the GTR model plus gamma correction and estimated the proportion of invariable sites. Both capsid and RdRp trees showed similar topological structures, presenting three AaTV clades (Figure 3). In addition, these trees have compatible evolutionary rates (see the branch length scaled in nucleotide substitutions per site). These clades were also identical to those observed in the network constructed with near-complete genomes (Figure 2). It is interesting to note that all sequences belonging to clades A, B, and C in the network were located in their respective clades in the capsid and RdRp trees, with the exception of the sequences MN053724, MN053725, and MN053732 that swapped between clade A and clade B. Specifically, AaTV sequences MN053725, and MN053732 belong to clade B in the network and in the capsid tree, while in the RdRp they are in clade A. On the other hand, AaTV MN053724 belongs to clade A in the network and in the capsid tree, while in the RdRp tree, it is located in clade B.

### 3.4. Recombination Signal in AaTV Sequences

The phylogenetic analysis showed that the sequences MN053724, MN053725, and MN053732 were in different clades in trees constructed with capsid and RdRp, and these findings suggest that these sequences have parts of their genomes composed of distinct segments originating from different phylogenetic clades. Thus, we decided to initially detect any signal of recombination in the alignment composed of near-complete AaTV sequences. AaTV MT435499 was excluded from the recombination analysis owing to the poor quality of this sequence. We used an approach (INTERVAL) that estimates site-by-site variations in recombination rates along the lengths of nucleotide sequence alignments. This can therefore potentially be used in a similar way to identify the recombination breakpoint density signal. The recombination rates (Rho values in the *y*-axis in Figure 4a) estimated using the alignment containing all AaTV sequences indicated a unique high recombination peak (indicated by a gray arrow in Figure 4a) close to the genome position of 3000 (*x*-axis in Figure 4a). As a control, we also estimated recombination rates in an alignment in which the sequences MN053724, MN053725, and MN053732 were excluded. The results indicated that the site-by-site recombination rates were lower (*y*-axis in Figure 4b) than rates estimated with all AaTV sequences (see the scale in the *y*-axis in Figure 4). The above results indicate clearly that MN053724, MN053725, and MN053732 are recombinants and have at least one breakpoint. In addition, this result indicates that all other AaTV sequences are non-recombinant.

### 3.5. Mosaic Patterns of Recombinant AaTV Sequences

Until now, we have confirmed that the sequences MN053724, MN053725, and MN053732 have chimeric genomes, likely composed of AaTV strains from clade A and clade B, and that they also have at least one breakpoint. Therefore, a very detailed analysis was performed in order to determine the number of recombination breakpoints and the mosaic patterns of the AaTV chimeric sequences. We used distinct methods to detect recombination, all of which confirmed the presence of one single breakpoint. All methods used detected the signal of one single recombination breakpoint in the sequences MN053724, MN053725, and MN053732 (Table 1). Notably, the locations of the recombination breakpoint coincided in all three chimeric sequences (Appendix A). 

To better characterize the mosaic patterns of the recombinant sequences, we used methods that identified parental sequences, which most likely contributed to the formation of chimeric sequences. The mosaic structure of MN053724 is composed of the capsid gene originating from the sequences of clade A and the RdRp gene originating from clade B (Figure 5a and Appendix A). Furthermore, sequences MN053725 and MN053732 have identical mosaic patterns, and their genomes are composed of the capsid from clade B and the RdRp gene from clade A (Figure 5a and Appendix A). Likewise, the location in the genome where there is a shift in the similarities between clades (indicated by red arrows in Figure 5a) coincides in all chimeric sequences. The recombination breakpoint detected in the AaTV chimeric sequences is located at position 2933 in an intergenic region between the capsid and RdRp genes. The intergenic region of AaTV contains 110 bp and is highly conserved among all sequences except at position 2933. Sequences that belong to clade A have C at position 2933, while all other sequences have T at this position.

### 3.6. RNA Structure in the AaTV Intergenic Region

Because the recombination breakpoint was found in the same location in the intergenic region of all AaTV chimeric strains, we decided to investigate whether this region contained a structure that could interfere with the RdRp enzyme’s processivity during genome replication. We used the complete intergenic region (110 bp) to predict the secondary RNA structure. The predicted structure shows a hairpin-like structure that is formed before position 2933 (indicated by an arrow in Figure 5b).

## 4. Discussion

Genetic recombination in RNA viruses results in the generation of chimeric RNA molecules that contain sequences from each parent, thus giving rise to new allele combinations [32]. RNA viruses are an important mechanism that increases genetic variability and adaptation [33]. The most conceivable mechanism of RNA recombination is ‘copy choice’ recombination. In this model, the RNA-dependent RNA polymerase changes from one RNA template (the donor template) to another (the acceptor template) during synthesis. It remains bound to the nascent nucleic acid strand, creating a mixed-ancestral RNA molecule [34]. The sequence similarity between the donor and acceptor template nucleic acid molecules is hypothesized to act as a cue to switch templates. As a result, RNA recombination is frequently “homologous,” as it happens in places with a lot of sequence similarity. However, the degree of local sequence identity between the RNA templates, transcription kinetics, and secondary RNA structure are some of the variables that affect template switching [35,36].

In this study, we show the grouping structure of AaTV lineages clustered into at least three phylogenetic clades. We also describe the mosaic genome of recombinant strains circulating in the *Aedes aegypti* population in Guadeloupe. Because these chimeric viruses were detected in the same geographical Caribbean region, it is plausible that they infect overlapping mosquito populations. In addition, the identification of three recombinant sequences that were sampled between 2016 and 2017 suggests that these chimeric lineages are currently circulating in the region. The impact of recombination on the adaptation to the diversity of AaTVs in natural populations needs to be further explored. It is known that Aedes populations can spread pathogenic viruses and function as reservoirs of emerging pathogens [6,7,37]. Insect-specific viruses (ISV) infect insects and insect cells but do not replicate in vertebrates or vertebrate cells [21]. It should be noted that, while these agents are referred to as “insect-specific viruses,” the vast majority were discovered in mosquitoes [7]. This is likely due to the fact that ISVs were primarily discovered in the context of arbovirus surveillance studies, which often preferentially target mosquitoes [38]. Many ISVs are found in viral families associated with arboviral diseases. Phylogenetic studies have provided some context for the evolution and ecology of these agents, indicating that ISVs from the Flaviviridae and Bunyaviridae families are ancient agents with divergent lineages, lending support to the hypothesis that these ISVs co-evolved and diversified with their insect hosts [39]. Based on these findings, it is likely that many pathogenic arboviruses acquired their dual-host diversification through adaptive evolution, which made an ancient ISV capable of infecting vertebrates [19,40]. For these reasons, ISVs, such as AaTVs, represent potential emerging pathogens, yet the effects and outcomes of these viruses in mosquitoes are poorly understood [6,21,41]. Furthermore, it has been proposed that ISVs may interfere with the infectivity of other arboviruses [40,42]. For example, *Culex flavivirus* [43,44], Palm Creek virus [45], and La Crosse virus [44] have been reported to alter mosquito susceptibility to certain arboviruses. Likewise, in *Aedes aegypti,* coinfection with the Zika virus increases the viral genome copies of the Guadeloupe mosquito virus and decreases the viral copies of the *Aedes aegypti* totivirus [40].

The identification of different chimeric AaTV strains all having a single breakpoint at the same position suggests that the non-coding intergenic regions are a recombination hotspot. The 5’ untranslated region (5’ UTR) of totiviruses is very structured and functions as an internal ribosome entry site (IRES). The noncoding intergenic region is highly conserved among AaTVs; it is also prone to the formation of a structure. It is plausible that the RNA structure predicted in the intergenic region of AaTV may affect the processivity of the RNA polymerase during replication. Thus, by stalling RNA polymerase, the RNA structure facilitates template switching and the synthesis of chimeric AaTV genomes. There is much empirical evidence that RNA secondary structures induce homologous recombination in viruses [46,47]. For example, the Turnip Crinkle virus (TCV) genome’s recombination hotspots have been shown to be localized near hairpin structures [48]. Similarly, a deep sequencing investigation of the breakpoints in the poliovirus genome and gene synthesis has shown that the majority of these breakpoints are connected to RNA secondary structures, such as hairpins and stem-loop structures [49].

## 5. Conclusions

Mosquitoes harbor a great variety of human pathogenic viruses, and they function as reservoirs of potential new pathogens. Here, we provide some examples of how AaTV can increase genomic diversity through recombination that forms chimeric genomes composed of genes from distinct clades. Thus, the role of insect viruses in the emergence of new arboviruses requires further investigation.

## Figures and Tables

**Figure 1 viruses-14-02467-f001:**
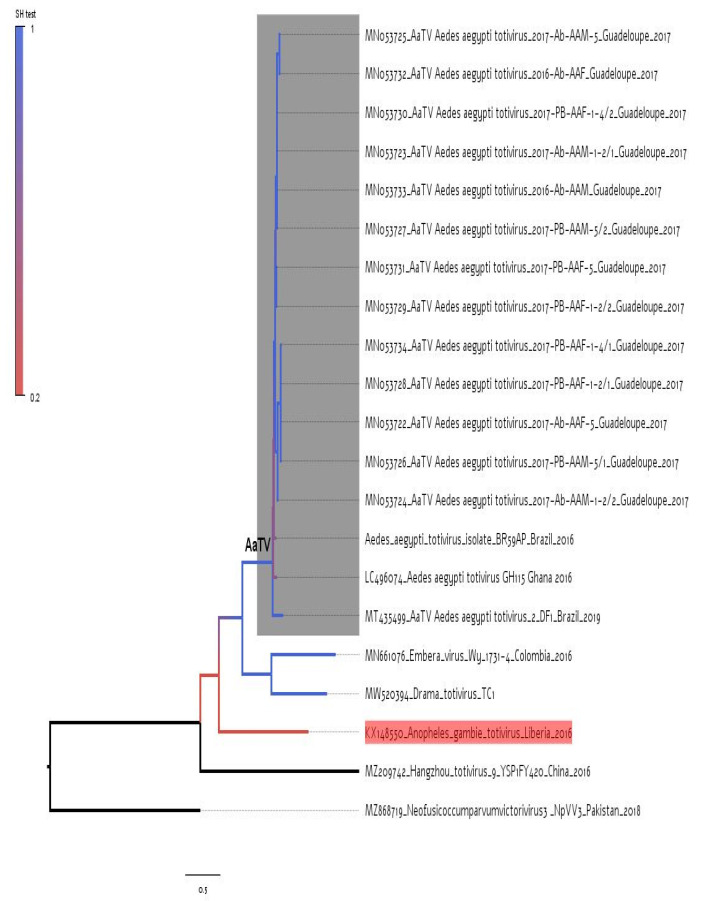
A maximum likelihood tree was constructed with near-complete genomes of totiviruses. The monophyletic group formed by totiviruses identified in *Aedes aegypti* mosquitoes (AaTV) is indicated by a gray area. The totivirus identified in an *Anopheles gambiae* mosquito is indicated by a blue arrow in the tree. Totiviruses identified in distinct hosts were also included in the tree as references. The tree was rooted with the reference MZ868719, which isolated the pathogenic plant fungi *Neofusicoccum parvum*. Branch support based on the Shimodaira–Hasegawa test is indicated by the color scale. The horizontal line under the tree is a scale that indicates the number of nucleotide substitutions per site.

**Figure 2 viruses-14-02467-f002:**
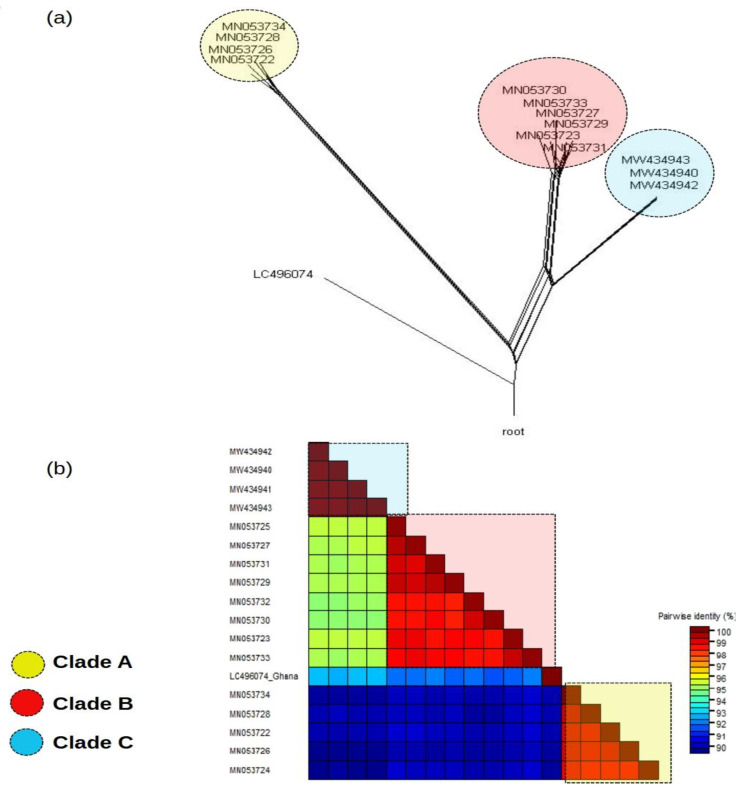
Phylogenetic clades of AaTVs. (**a**) The phylogenetic network constructed with the near-complete genomes of AaTV. The network shows three clades (delineated by colored circles) composed of AaTV sequences. (**b**) Similarity matrix calculated with the near-complete genome of the AaTVs. The identity score is indicated by a colored scale. Colored rectangles delineate sequences within a certain clade. The clades A, B, and C of the network are connected by dashed lines in the matrix.

**Figure 3 viruses-14-02467-f003:**
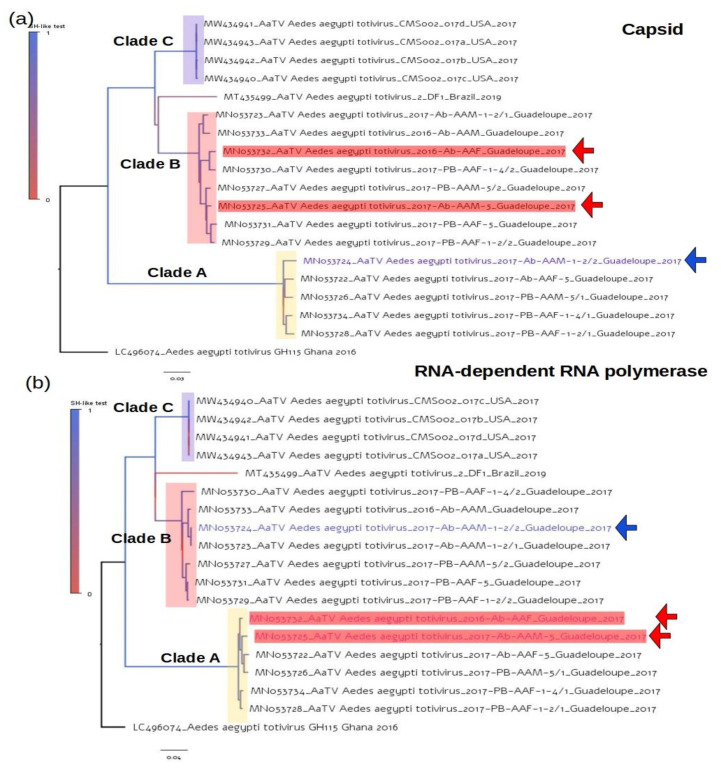
Phylogenetic Capsid and RNA-dependent RNA-polymerase trees. (**a**) Phylogenetic tree constructed from the AaTV capsid gene region. (**b**) AaTV phylogenetic tree built with the RdRp gene region. The maximum likelihood criterion was used to build both trees, assuming the GTR model with gamma correction and the estimated proportion of invariable sites. The Shimodaira–Hasegawa test was used to determine branch support, which is represented by a color scale. Phyloclades are named and represented in the tree by colored rectangles. The horizontal bar beneath each tree represents the number of nucleotide substitutions per site.

**Figure 4 viruses-14-02467-f004:**
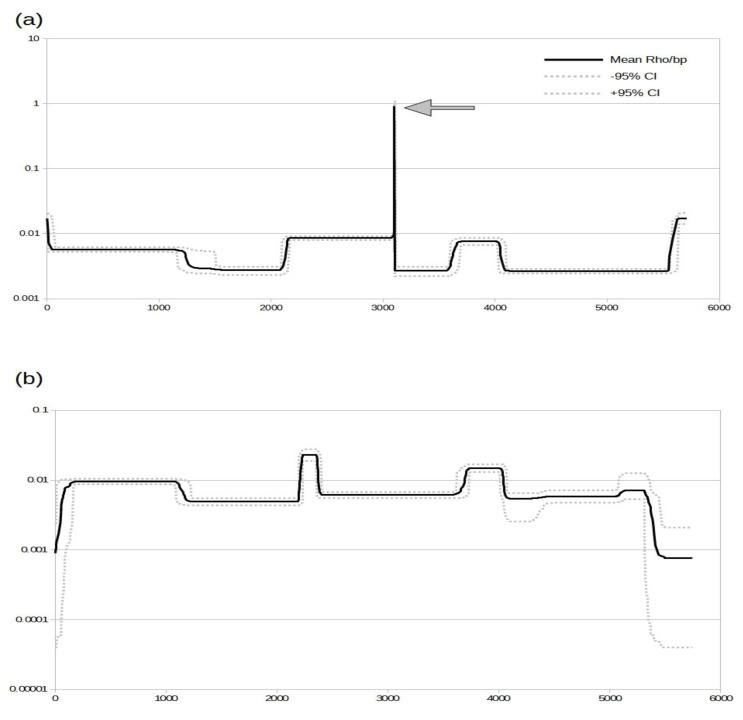
Recombination rates per site in the AaTV genome. (**a**) Recombination rates estimated in near-complete AaTV sequences. (**b**) Recombination rates estimated in an alignment excluding the sequences MN053724, MN053725, and MN053732. The *x*-axis indicates the genome position. The *y*-axis indicates the recombination rates (Rho) estimated site-by-site. The *y*-axis is shown in the log scale. Black lines indicate mean rho per bp, and dashed gray lines indicate 95% confidence intervals.

**Figure 5 viruses-14-02467-f005:**
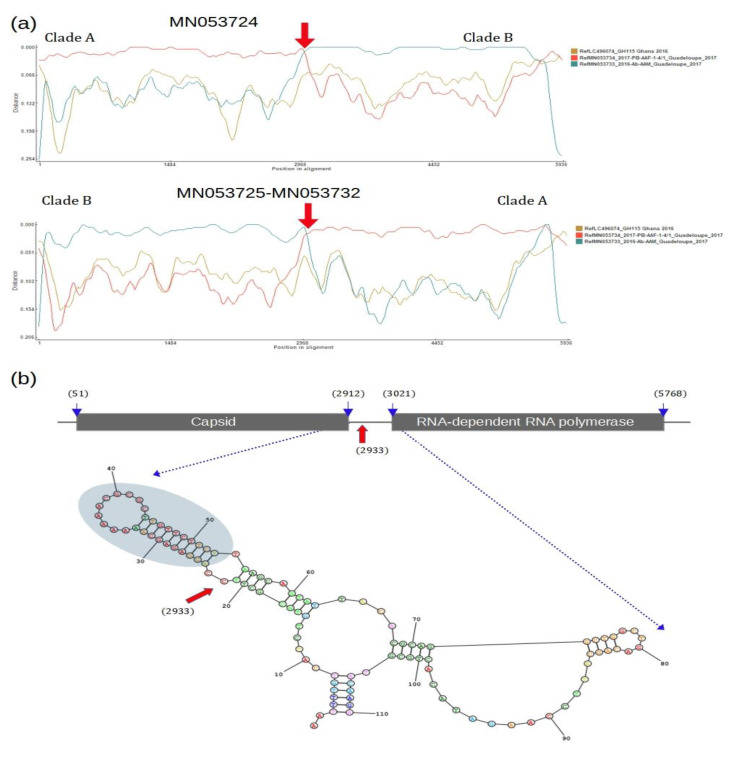
A genomic map of chimeric AaTVs. (**a**) Similarity plot of MN053724 (upper panel) and MN053725 and MN053732 (low panel). The *x*-axis shows the alignment position. The *y*-axis shows the distance between the query sequence and the parental sequences. The colored lines indicate the similarity along the genomic regions of query sequences. (**b**) Diagram illustrating the location of capsid and RdRp genes in the AaTV genome. Numbers inside parentheses indicate the start and end of the genes. The red arrow in the intergenic regions indicates the location of the recombination breakpoint in the chimeric AaTV sequences. Dashed arrows show the hypothetical RNA structure formed in the intergenic region. The structure with favorable free energy (with a minimum free energy of −28.10 kcal/mol) is indicated by a gray circle.

**Table 1 viruses-14-02467-t001:** Approximate *p*-values of recombination signal in the AaTV genome.

Sequence ID	Recombination Method								
	GENECONV	LARD	DSS	Bootscan	Chimaera	SiSscan	Maxchi	3Seq	RDP
MN053724	3.03 × 10^−62^	1.42 × 10^−114^	6.29 × 10^−35^	2.15 × 10^−77^	6.60 × 10^−38^	2.71 × 10^−45^	3.95 × 10^−38^	1.51 × 10^−101^	6.25 × 10^−95^
MN053725 MN053732	3.00 × 10^−57^	1.02 × 10^−107^	3.58 × 10^−48^	5.40 × 10^−63^	2.80 × 10^−37^	3.93 × 10^−42^	1.37 × 10^−35^	3.04 × 10^−118^	2.71 × 10^−80^

RDP: the *p*-value is calculated by Binomial distribution. GENCONV: the *p*-value is calculated using Blast-Like Karlin–Altschul and permutation test. Bootscan: Bootstrapping, binomial distribution, and χ^2^ distribution. MaxChi and Chimaera: χ^2^ distribution and permutation. SISscan: Permutation and Z-test. LARD: Likelihood ratio test. 3Seq: Exact test. DSS: the *p*-value is calculated using parametric bootstrap.

## Data Availability

Not applicable.

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
