# Peer review of "Is the Intergenic Region of *Aedes aegypti* Totivirus a Recombination Hotspot?"

_viruses, 2022, doi:10.3390/v14112467_

Round 1
Reviewer 1 Report
The manuscript entitled “Is the intergenic region of Aedes aegypti Totivirus a 2 recombination hotspot?” reports bioinformatics analysis of 25 Aedes aegypti totivirus genome sequences. By comparison of sequences and further phylogenetic analysis, authors claim the identification of tentative recombination position in a genomes of three AaTV isolates.
The manuscript has several significant shortages to be accepted for the publication. The Introduction is full of misstatements (see below for details), unfortunately undescriptive and, in addition, written in flawed English. The M&M part misses section 2.3, while section 2.4 is overwhelmed by irrelevant information. While the Results and Discussion sections are of sufficient quality to formulate the outcome(s), the statement in Conclusions (Lane 319) is both too observational for such an analysis and too inconclusive given the unilateral type of experimental approach involved.
Minor points:
Lane 14: structural organization of totiviruses is not the same along all representatives of the family.
Lane 30: “Totiviridae is……………..virus family”. The name is not a family.
Lane 34: sentence appears contradictory to the statement of the next sentence.
Lanes 35-37: reference is needed.
Lane 37: statement not true for the viruses analyzed in this Manuscript.
Lanes 41, 42: reference [1] is not proper here.
Lanes 43-46: statement is simply wrong, reference is missing.
Lane 60: reference 5 is wrong number or missing.
Lane 74: Title is misleading for “…genetic analysis”.
Lanes 74-89: section 2.3 is missing.
Lane 89: detailed description of all linked algorithms is irrelevant here.
Major points:
Lanes 256-258: what’s the deltaG value for this structure? This may shed light if the position is targeted for recombination indeed.
Lanes 300-301: statement “…is not a coincidence” is too bold without further proofing. Formulations alike “It is plausible that….may affect…” (lanes 305-306) involved in the description of a principal connection between the sequence analysis and the genome structures of (only) three isolates reads as a bad joke. It cannot be viewed or treated as an evidence nor the proof, while might be a good hypothesis for the further investigation.
Lane 319: conclusion, which appears central in this Manuscript, is written in flawed grammar. In addition, it is rather general and of low value to the readers.
Author Response
The manuscript entitled “Is the intergenic region of Aedes aegypti Totivirus a 2 recombination hotspot?” reports bioinformatics analysis of 25 Aedes aegypti totivirus genome sequences. By comparison of sequences and further phylogenetic analysis, authors claim the identification of tentative recombination position in a genomes of three AaTV isolates. The manuscript has several significant shortages to be accepted for the publication. The Introduction is full of misstatements (see below for details), unfortunately undescriptive and, in addition, written in flawed English. The M&M part misses section 2.3, while section 2.4 is overwhelmed by irrelevant information. While the Results and Discussion sections are of sufficient quality to formulate the outcome(s), the statement in Conclusions (Lane 319) is both too observational for such an analysis and too inconclusive given the unilateral type of experimental approach involved. Resp: We changed the introduction in order to clarify that we are describing recombination in Aedes aegypti totivirus-like, which is an unclassified member of the genus Totivirus. We also removed the irrelevant description of recombination methods from the M&M section. Besides, the conclusion was also changed. Minor points: Lane 14: structural organization of totiviruses is not the same along all representatives of the family. Resp: The introduction section was modified in the new version of the manuscript. Lane 30: “Totiviridae is……………..virus family”. The name is not a family. Resp: This was corrected in the revised manuscript. Lane 34: sentence appears contradictory to the statement of the next sentence. Resp: This sentence has been removed from the revised manuscript. Lanes 35-37: reference is needed. Resp: All references were updated in the new version of the manuscript. Lane 37: statement not true for the viruses analyzed in this Manuscript. Resp: This statement has been removed from the introduction section. Lanes 41, 42: reference [1] is not proper here. Resp: This reference was removed. Lanes 43-46: statement is simply wrong, reference is missing. Resp: The introduction section was modified. Lane 60: reference 5 is wrong number or missing. Resp: References were updated in the revised manuscript. Lane 74: Title is misleading for “…genetic analysis”. Resp: Topic 2.2 was renamed to ‘Alignment and phylogenetic analysis Lanes 74-89: section 2.3 is missing. Resp: subsections in the M&M were renamed: 2.1. Dataset; 2.2. Alignment and Phylogenetic analysis; 2.3. Recombination Signal and 2.4. RNA Secondary structure prediction Lane 89: detailed description of all linked algorithms is irrelevant here. Resp: This description was removed from the M&M section. Major points: Lanes 256-258: what’s the deltaG value for this structure? This may shed light if the position is targeted for recombination indeed. Resp: The hypothetical structure has minimum free energy of -28.10 kcal/mol. This was included in the manuscript. Lanes 300-301: statement “…is not a coincidence” is too bold without further proofing. Formulations alike “It is plausible that….may affect…” (lanes 305-306) involved in the description of a principal connection between the sequence analysis and the genome structures of (only) three isolates reads as a bad joke. It cannot be viewed or treated as an evidence nor the proof, while might be a good hypothesis for the further investigation. Resp: We changed the sentence to: ‘The identification of different chimeric AaTV strains all having a single breakpoint at the same position suggests that the non-coding intergenic regions are a recombination hotspot.’ Lane 319: conclusion, which appears central in this Manuscript, is written in flawed grammar. In addition, it is rather general and of low value to the readers. Resp: We also changed the conclusion in the revised manuscript.Reviewer 2 Report
In this study, the authors identified recombinant genomes amongst isolates of toti-like viriuses from different populations of Aedes aegypti mosquitoes. Assessing the relationships of these viruses to each other revealed some incongruence detected in gene-specific phylogenies (based on either RdRP or capsid sequences), and the authors discovered that this incongruence was caused by recombination occurring at an intergenic region between the capsid and RdRP ORFs. The authors identified a secondary structure in this region that likely could contribute to template-switching by the RdRP, which would account for the occurrence of recombinant genomes amongst these viruses.
The authors provide an abundance of evidence from several analytical methods to demonstrate the presence of recombination amongst the viruses they surveyed and have written an interesting discussion of the results. One issue is that the text in some of the figures is very small and I had to zoom in as much as 400% to read it; it would be good if the authors could do something to improve the readability of the text in their figures. Also, I think it would be good to make Table S1 into a regular table incorporated in the paper, as it highlights the strength of the evidence for recombination occurring amongst these viruses.
Other comments:
11) Lines 2, 11, 69, 72: The genus name “Totivirus” should be capitalized, while “totivirus” when used to refer to the virus should not be capitalized unless at the beginning of a sentence.
22) Lines 41-42: Some English copyediting is required; for example, I think the sentence here probably was intended to say that totiviruses produce a fusion protein containing RdRP and capsid sequences and that the RdRp is packaged in the virion.
33) Lines 43-46: Can the authors clarify whether any of the mosquito toti-like viruses have been officially classified by the ICTV? There do not appear to be any mosquito totivirus species currently in the ICTV list.
44) Figure 1: Does the fungal totivirus in this phylogeny correspond to an officially classified totivirus species?
55) Figure 2: It looks like this analysis is missing the Brazilian isolate (MT435499). The subsequent Figure 3 indicates that it is not a part of the clades A, B, or C; perhaps some text can be added to indicate that.
66) Lines 257-258, Figure 5b: What is the deltaG of this structure?
77) Line 302: Re-write as “…likely is not a coincidence.”
Author Response
In this study, the authors identified recombinant genomes amongst isolates of toti-like viriuses from different populations of Aedes aegypti mosquitoes. Assessing the relationships of these viruses to each other revealed some incongruence detected in gene-specific phylogenies (based on either RdRP or capsid sequences), and the authors discovered that this incongruence was caused by recombination occurring at an intergenic region between the capsid and RdRP ORFs. The authors identified a secondary structure in this region that likely could contribute to template-switching by the RdRP, which would account for the occurrence of recombinant genomes amongst these viruses. The authors provide an abundance of evidence from several analytical methods to demonstrate the presence of recombination amongst the viruses they surveyed and have written an interesting discussion of the results. One issue is that the text in some of the figures is very small and I had to zoom in as much as 400% to read it; it would be good if the authors could do something to improve the readability of the text in their figures. Also, I think it would be good to make Table S1 into a regular table incorporated in the paper, as it highlights the strength of the evidence for recombination occurring amongst these viruses. Resp: We included higher resolution figures (zipped files). Table S1 was inserted in the main text (Table 1). Other comments: 11) Lines 2, 11, 69, 72: The genus name “Totivirus” should be capitalized, while “totivirus” when used to refer to the virus should not be capitalized unless at the beginning of a sentence. Resp: The introduction section was changed in the new version of the manuscript and viral genera and families were capitalized accordingly. 22) Lines 41-42: Some English copyediting is required; for example, I think the sentence here probably was intended to say that totiviruses produce a fusion protein containing RdRP and capsid sequences and that the RdRp is packaged in the virion. Resp: This sentence was changed in the new version of the manuscript. 33) Lines 43-46: Can the authors clarify whether any of the mosquito toti-like viruses have been officially classified by the ICTV? There do not appear to be any mosquito totivirus species currently in the ICTV list. Resp: These viruses have been tentatively placed in the Totiviridae family and have not been definitively classified. 44) Figure 1: Does the fungal totivirus in this phylogeny correspond to an officially classified totivirus species? Resp: The reference Neofusicoccum parvum victorivirus 3 isolate NpVV3 is classified in the genera Victorivirus 55) Figure 2: It looks like this analysis is missing the Brazilian isolate (MT435499). The subsequent Figure 3 indicates that it is not a part of the clades A, B, or C; perhaps some text can be added to indicate that. Resp: The sequence MT435499 was excluded from the recombination analysis because of the poor quality of this sequence. It has an unusually long 5’ UTR (1318bp), the Capsid region has many unique substitutions and the end region of RdRp is incomplete. 66) Lines 257-258, Figure 5b: What is the deltaG of this structure? Resp: The hypothetical structure has minimum free energy of -28.10 kcal/mol. This was included in the manuscript. 77) Line 302: Re-write as “…likely is not a coincidence.” Resp: This sentence was changed in the revised manuscript.